# Molecular Screening for High-Risk Human Papillomaviruses in Patients with Periodontitis

**DOI:** 10.3390/v15030809

**Published:** 2023-03-22

**Authors:** Kalina Shishkova, Raina Gergova, Elena Tasheva, Stoyan Shishkov, Ivo Sirakov

**Affiliations:** 1Laboratory of Virology, Faculty of Biology, University of Sofia “St. Kl. Ohridski”, 8 Dragan Tzankov Blvd., 1164 Sofia, Bulgaria; k_kostova1@abv.bg (K.S.);; 2Department of Medical Microbiology, Faculty of Medicine, Medical University of Sofia, 2 “Zdrave” Str., 1431 Sofia, Bulgaria; 3G-Lab Ltd., 2 “Hristo Belchev”, 1000 Sofia, Bulgaria; 4Department of Zoology and Anthropology, Faculty of Biology, Sofia University “St. Kliment Ohridski”, 8 Dragan Tzankov Blvd., 1164 Sofia, Bulgaria

**Keywords:** high-risk HPV, bacteria, periodontitis

## Abstract

Members of the *Papillomaviridae* family account for 27.9–30% of all infectious agents associated with human cancer. The aim of our study was to investigate the presence of high-risk HPV (human papilloma virus) genotypes in patients with periodontitis and a pronounced clinical picture. To achieve this goal, after proving the bacterial etiology of periodontitis, the samples positive for bacteria were examined for the presence of HPV. The genotype of HPV is also determined in samples with the presence of the virus proven by PCR (polymerase chain reaction). All positive tests for bacteria associated with the development of periodontitis indicated the presence of HPV. There was a statistically significant difference in HPV positive results between the periodontitis positive target group and the control group. The higher presence of high-risk HPV genotypes in the target group, which was also positive for the presence of periodontitis-causing bacteria, has been proven. A statistically significant relationship was established between the presence of periodontitis-causing bacteria and high-risk strains of HPV. The most common HPV genotype that tests positive for bacteria associated with the development of periodontitis is HPV58.

## 1. Introduction

Representatives of the *Papillomaviridae* family make up 27.9–30% of all infectious agents associated with human cancer [1]. Although papillomaviruses were poorly studied in the 1950s and 1960s, this period is associated with some important advances, including physicochemical analysis of the virion and the demonstration that HPV replication is associated with the process of differentiation of the infected epithelium [2]. The advent of molecular cloning in the 1970s initiated more extensive research and provided opportunities for its mapping, sequencing, and in-depth study of the structure and functions of viruses [3]. Bovine papillomavirus type 1 (BPV1) is a standard model for conducting these studies because the virus induces focal transformation of established rodent cell lines [4,5]. In addition, papillomaviruses are an important problem in veterinary medicine, causing papillomas in dogs, cats, and cattle. Further, bovine papillomaviruses 1 (BPV1) infect Equidae, leading to the development of sarcoids [6]. There is evidence of interspecific transmission of this virus from cattle to humans and the development of colorectal tumors as a result of red meat consumption [7]. Although the study of animal papillomaviruses continues to provide new information, the importance of PV for human medicine has shifted the focus to HPV analysis, especially when it was found that the biochemical properties of some nonstructural viral proteins differ from those of BPV1 [8,9].

It is estimated that about 16% of the world’s population is infected with HPV. According to a study by K.J. Syrjanen, nearly 80% of women are at risk for HPV infection. This is not the case with regard to affecting the oral epithelium; for example, in the United States, it has been found that 7% of people aged 14–69 years are infected with HPV, and the infection in men is almost three times more common [3]. The majority of HPV infections in men are thought to be transitory and asymptomatic, with no clinical manifestations [10]. A pioneer in the study of HPV in Bulgaria is Prof. Zl. Kalvachev, and a screening has been conducted by a number of researchers [11,12,13]. Papillomaviruses have oncogenic potential and are associated with the development of malignancies in humans and animals. There is a direct link between HPV types 16 and 18 and the development of cervical cancer in women [14].

Oncogenic serotypes of human papillomaviruses are used as a tumor marker associated with cervical cancer due to their safe participation in the etiology of this cancer, and in many cases, they are responsible for the development of neoplastic lesions of the penis, tongue, and more rarely of mucosal lesions with other localization. An Italian team of authors conducted a study that reports that women (20.4%) with a diagnosis of cervical-HPV and their male partners (30.7%) are at high risk for subclinical oral HPV infection [15].

The presence of HPV in the oral cavity has been associated with various lesions, but the presence of viruses has also been found without clinical manifestation [16]. The link and relationship between prokaryotes and viruses has been proven, with some bacteria supporting the development of infections with certain viruses—directly and indirectly [17,18,19]. The incidence of human papillomavirus (HPV)-related oropharyngeal squamous cell carcinomas is increasing globally. Common oral conditions such as periodontitis may contribute [20]. It was reported that the periodontal pathogens Fusobacterium, Prevotella, and Alloprevotella were enriched but commensal Streptococcus depleted in oral cavity squamous cell cancer [21]. The presence of high-risk human papillomaviruses in periodontal pockets patients of diagnosed with chronic periodontitis but not suffering from head and neck squamous cell carcinoma in the present day could link periodontitis to HPV-related squamous cell carcinoma. Prevalence studies detecting the presence of HPV in marginal periodontium as well as prospective studies of HPV-positive periodontitis patients are required to explore this possible link [22]. The aim of our study was to prove the relationship between periodontitis-causing bacteria and the presence of high-risk HPV genotypes in patients from Bulgaria. Patients have a pronounced clinical picture of periodontitis in the absence of oropharyngeal lesions in the oral cavity. In contrast to previous studies, our genotyping found that the most common genotype in samples positive for periodontitis bacteria was HPV type 58, followed by types 16, 33, and 35. 

## 2. Materials and Methods

### 2.1. Samples

Two hundred and thirty samples were used from patients with clinical manifestations of gingivitis and periodontitis without a pronounced clinical picture of oropharyngeal lesions in the oral cavity (target group). The age of the target group patients ranges from 25 to 59 years. All of them required periodontal procedures and placement of implants. There is no evidence in the target group of the presence of immunosuppressed individuals. Samples for the presence of bacteria causing periodontitis were taken only from the gingival margin due to the specificity of the disease. Samples were taken using 0.5 mm dental paper pins using 5 pins per patient. For a control group, 460 buccal mucosa and coronal margin samples from 230 people without clinical signs of periodontitis were used. The representatives of the control group were of a similar age: from 30 to 60 years old. They were screened for high-risk HPV. We used sterile brushes (Biomed Future, Bulgaria) for the buccal mucosa samples and thin, metallic, sterile swabs (Biomed Future, Bulgaria) for those from the gingival margin. 

### 2.2. Primer Systems

Various primer systems were used in the present work to detect genomic fragments of HPV. Consensus primers MY09/11 cover a region of the DNA of the virus with a size of 450 nucleotides (nt) encoding the L2 and L1 proteins and the triple primers GPE—the genes encoding proteins E6 and E7 [23] of different sizes depending on the genotype. Primers for the beta globin gene were used as an internal control of the system. 

### 2.3. DNA Isolation and In-Vitro Diagnostic Kits

An ISOLATE II DNA Mini kit with a silica membrane was used to isolate the DNA (Bioline, Meridian Bioscience, Memphis, TN, USA). The isolation protocol was according to the manufacturer’s instructions.

The pathogenic bacteria associated with periodontitis were detected by micro-IDent^®^plus11 kit (Hain Lifescience GmbH, Nehren, Germany). Reactions were performed according to the manufacturer’s instructions using a FluoroCycler^®^ 12 PCR apparatus (Hain Lifescience GmbH, Nehren, Germany), and a TwinCubator (Hain Lifescience GmbH, Nehren, Germany) was used to perform the hybridization. 

An Amplisens kit (Ecoli Dx, s.r.o., Prague, Czech Republic) was used to detect HPV types 16, 18, 31, 33, 35, 39, 45, 52, 56, 58, 59, and 66. The protocols were according to the manufacturer’s instructions. 

### 2.4. Polymerase Chain Reaction 

The commercial in vitro diagnostic kits and primers described in Table 1 (following a compliance check with the *GenBank*—National Center for Biotechnology Information, USA) were used to perform the reactions. Minor changes were made to the reaction protocols according to the melting temperature of the primers. To determine the temperature range of the primers, a gradient PCR was performed at the following temperatures: 51.0, 52.9, 54.4, 56.2, 58.4, and 60.3 °C. MayTaq HS mix (Bioline, Meridian Bioscience, Memphis, TN, USA) was used for PCR.

Primers PC04 and GH20 for the beta globin molecule gene (Saiki et al. 1986) forming a fragment with a size of 268 base pairs (bp) were used as internal control of the system. 

Consensus primers MY09-MY11 (10 pmol/µL) adopted as the gold standard were used to detect HPV, yielding a 450 bp fragment [19].

Consensus primers covering E6/E7 genes were also used to detect HPV: GP-E6-3F, GP-E6-5B, and GP-E6-6B primers yielding a 650–700 bp fragment depending on of the genotype.

The parameters of the reactions for proving HPV were as follows: initial denaturation—3 min, 40 cycles: denaturation 95 °C—45 s, connection 55 °C—45 s, extension 72 °C—45 s, final extension 72 °C—7 min, and storage 4 °C. 

### 2.5. Gel Electrophoresis

Gel electrophoresis was used for qualitative analysis of the extracted DNA and PCR products. Gel electrophoresis was performed with 2% agarose (Lonza Group AG, Basel, Switzerland), 10 ng/mL of ethidium bromide (Sigma-Aldrich, Merck KGaA, Saint Louis, MO, USA), 1 × TAE buffer, 100 bp DNA Ladder (New England Biolabs, Ipswich, MA, USA), and 1 kb DNA ladder (Bioline, Meridian Bioscience, Memphis, TN, USA) at 120–150 V and 70–120 mA for 30 min. 

### 2.6. Statistical Processing

For statistical processing of the results, chi-square test and McNemar’s test were used to determine the level of significance.

## 3. Results

DNA obtained from all tested samples had a purity greater than 1000 and an appropriate concentration greater than 20 ng/µL. 

Evidence of pathogenic bacteria associated with the development of periodontitis.

After DNA preparation, PCR and hybridization were performed to detect 11 pathogens causing periodontitis. The results showed that all examined samples from the target group (with a clinical picture of periodontitis) showed the presence of periodontitis bacteria. The presence of all tested bacteria was demonstrated, with several types of bacteria present in each sample. In all patients, the presence of *Pm* (*Peptostreptococcus micros*), *Fn* (*Fusobacterium nucleatum/periodo*) *En* (*Eubacterium nodatum*), and *Cr* (*Campylobacter rectus*) was recorded. A total of 230 samples from the coronary margin, isolated from the control group, were examined for the presence of the relevant bacterial pathogens. Samples for the presence of bacteria were taken only from the gingival margin due to the specificity of periodontal disease. After accounting for hybridization, 110 of the samples tested positive for periodontitis-causing bacteria. 

### 3.1. Polymerase Chain Reaction

After gradient PCR to determine the binding range of the beta globin primers PC04 and GH20, we found that at all temperatures tested (51.0–60.3 °C), there was a product of size 268 nt, and only at temperatures of 51.0 °C were there non-specific bands below 200 and below 100 nt due to the lower annealing temperature of the primers. 

The periodontitis-positive 230 samples from the target group were also probed with consensus primers GPE6/5B/6B and primers MY09/11. After the PCR, the presence of specific fragments with a size of about 700 bp and 450 bp was confirmed, which proves the presence of HPV. The results of the conducted reaction are presented in Figure 1.

Samples from the control group were probed with consensus primers MY09/11 and primers GPE6/5B/6B. Samples were taken from the gingival margin and buccal mucosa. The presence of specific fragments of 450 bp and about 700 bp was detected, which proved the presence of HPV. The number of HPV-positive samples taken from the gingival margin is higher, which is probably due to the structure of the oral epithelium of the gingiva, which allows the development of productive HPV infection with the assembly of functional virions, in contrast to the buccal mucosa. 

All HPV-positive samples from the target group were tested with an in vitro diagnostic kit for 12 high-risk HPV genotypes. After conducting PCR and visualization by agarose gel electrophoresis, the presence of fragments with a size characteristic of the respective genotypes was reported. The results of the studies are combined and presented in Table 2, Table 3 and Table 4. 

When examining the samples, it was found that in 110 of them, the presence of more than one HPV genotype was reported. High-risk genotypes 66, 31, and 59 were not reported in any of the samples.

In a screening of 230 coronal samples taken from the control group, 100 samples tested positive for HPV with the consensus primers. When tested for 12 high-risk HPV genotypes, 9 were positive for type 45 and 9 for type 18. The results are summarized in Table 3.

All HPV-positive samples (100) taken from the gingival margin of patients without clinical signs of periodontitis were also positive for periodontitis bacteria. Of 130 samples negative for HPV, 10 were positive for periodontitis bacteria. The presence of HPV and the absence of periodontitis-causing bacteria were not detected in any of the samples from the gingival margin of the control group.

When examining buccal mucosa samples from the control group with the consensus primers, 64 positive samples were found. Genotype 18 was proven in 16 of the samples. The results are presented in Table 4.

### 3.2. Statistical Processing of the Results

In the target group, in 170 patients or 73.9% of them, the presence of high-risk HPV genotypes was demonstrated, and all of them were positive for periodontitis bacteria. The results are presented graphically in Figure 2.

Eighteen genotypes positive for high-risk HPV, i.e., 7.8%, were found in the coronal margin samples in the control group. A statistically significant difference was found for this indicator (χ^2^ = 205; *p* < 0.001). The number of genotypes is also different when comparing samples taken from the coronal edge of the representatives of the target group and the control group—nine in the target group and only two genotypes in the control group for samples taken from the coronal edge.

A comparison of the proportion of samples from patients with high-risk HPV genotypes from the target group with the proportion of samples from patients with high-risk HPV genotypes from the control in which bacteria were also isolated from the coronal margin (16.4%) also showed that there was statistical significance (χ^2^ = 97.4; *p* < 0.001). The results are presented in Figure 3.

In the analysis of gingival margin samples in the control group, an association was found between the presence of bacteria and HPV. HPV was also detected in most patients (90.9%) positive for periodontitis-causing bacteria, while in patients without bacteria, there was not a single case with proven presence of HPV. The results were confirmed by the conducted McNemar’s test—χ^2^ = 8.1; *p* = 0.004.

Compared with the coronal margin results in the control buccal mucosa samples, only one high-risk HPV genotype was observed in fewer patients (7%). The observed difference is implausible.

## 4. Discussion

From all the research performed and the statistical processing of the results, we can conclude that there is a relationship between the presence of periodontitis-causing bacteria in patients with a pronounced clinical picture and high-risk HPV genotypes.

In contrast with colleagues [16] who found HPV in 22.9% of samples with neoplastic changes in the oral cavity, all samples tested by us that were positive for bacteria associated with the development of periodontitis were positive for HPV in all patients from the target group that was absent from the clinical picture of neoplastic changes in the oral cavity. 

A statistically significant difference was also found between the results obtained when examining the samples from the target group and the samples taken from the gingival margin of the control group in terms of HPV infection. In contrast to previous studies, our genotyping found that the most common genotype in samples positive for periodontitis bacteria was HPV type 58, followed by types 16, 33, and 35. HPV type 58 was found to be dominant in our study and was not associated with the development of papilloma lesions in the maxillofacial area, while the other widely represented genotypes 16, 33, and 35 were associated with the development of various papilloma lesions [16]. It is noteworthy that in patients with a pronounced periodontitis clinical picture and positive for the bacteria causing the disease, high-risk HPV genotypes predominate.

In the control group, a preponderance of HPV-positive samples was found in materials from the coronal margin compared to those from the buccal mucosa. This may be due to the fact that, theoretically, the structure of the oral gingival epithelium allows the development of a productive HPV infection with the assembly of functional virions, unlike the buccal mucosa. In relation to the structure of the gum and the pathological processes caused by the bacteria causing periodontitis, it is possible that there is a synergy between the bacteria and HPV. This interaction may be based on the mechanical exposure of the coronal margin (often accompanied by bleeding and pockets), which allows the virus to come into contact with one or both basement membranes as well as the junctional epithelium. We assume this is preparation for an opportunistic infection. Since the gingival mucosa bordering the tooth enamel lacks this differentiation, as does the oral epithelium or skin, i.e., keratinocytes [27], the absence of a clinical picture caused by HPV in the patients examined for periodontitis bacteria can also be explained since the viruses create productive inflammation only in the keratinocytes of the mucous membranes and skin [28]. In this regard, infection of the buccal epithelium should be non-productive, which could explain the higher number of positive samples when examining materials from the coronal margin compared to those from the buccal mucosa in samples from the control group. Furthermore, there is no evidence in the literature that the HPV types we found in the periodontitis-positive study group are associated with gingival lesions.

## 5. Conclusions

HPV localizes to the inflamed periodontal tissue and is thought to infect the basal keratinocytes in the ulcerated epithelium of the gingival sulcus. Inflammatory periodontal pockets play an important role as a reservoir for HPV. Although interactions between HPV and periodontal bacteria remain unclear, oral HPV infection may be associated with a characteristic oral microbiome. Carcinogenic HPV and periodontitis are likely to contribute to the development of oral cancer. To date, however, oral expression of HPV E6/E7 (transcriptionally active HPV) has not been fully studied in individuals with periodontitis. Overall, the available literature suggests that oral HPV may be associated with periodontitis. To clarify the relationship between oral HPV and periodontitis, the effects of clinical factors contributing to the spread of HPV DNA in the mouth must be considered [29]. 

In our experiments, the presence of all tested bacteria was detected, with several types of bacteria present in each sample. The presence of *Pm* (*Peptostreptococcus micros*), *Fn* (*Fusobacteric nucleatum/periodo*), *En* (*Eubacterium nodatum*), and *Cr* (*Campylobacter rectus*) was recorded in all patients. No statistically significant association was found between well-defined periodontitis-causative agents and well-defined high-risk HPV genotypes. We have reason to believe that it is an opportunistic infection due to the specificity of the disease and the replication cycle of HPV.

In the studies we conducted in Bulgaria, we found that for patients with a clinical picture of periodontitis and the presence of bacteria causing the corresponding disease, in the absence of a clinic of oropharyngeal lesions in the oral cavity, the presence of high-risk HPV was proven with a statistically reliable frequency. In control samples from the buccal mucosa and periodontal margin, the presence of high-risk HPV genotypes was significantly less. We can conclude that there is a relationship between the presence of periodontitis bacteria in patients with a pronounced clinical picture and high-risk HPV genotypes as well as a relationship between the presence of these bacteria and the presence of HPV. We could conclude that oral HPV infection is probably an opportunistic infection of the inflamed and bacterially infected gingival tissues. The presence of HPV in patients with periodontitis may influence disease severity and response to treatment. Therefore, the identification of HPV in patients with periodontitis may be essential to determine the appropriate course of treatment.

## Figures and Tables

**Figure 1 viruses-15-00809-f001:**
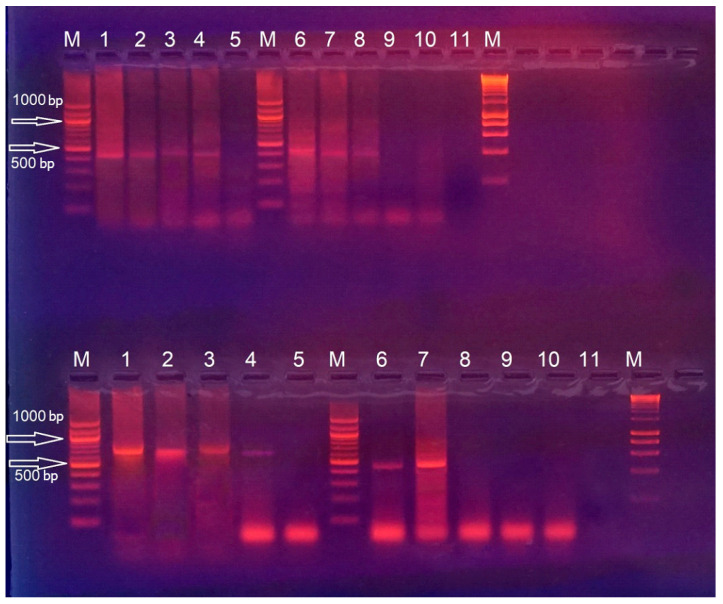
PCR with consensus primers GPE6/5B/6B and MY09/11of DNA materials from samples positive for periodontitis bacteria. M, 100 bp DNA marker; (**Upper row**) PCR performed with primers MY09/11. Reported fragment of about 450 bp; (**Bottom row**) PCR performed with primers GPE6/5B/6B. Reported fragment of about 700 bp; 10, negative control with DNA negative for HPV; M, 1 kb DNA marker. Samples from the target group were applied to the remaining starts.

**Figure 2 viruses-15-00809-f002:**
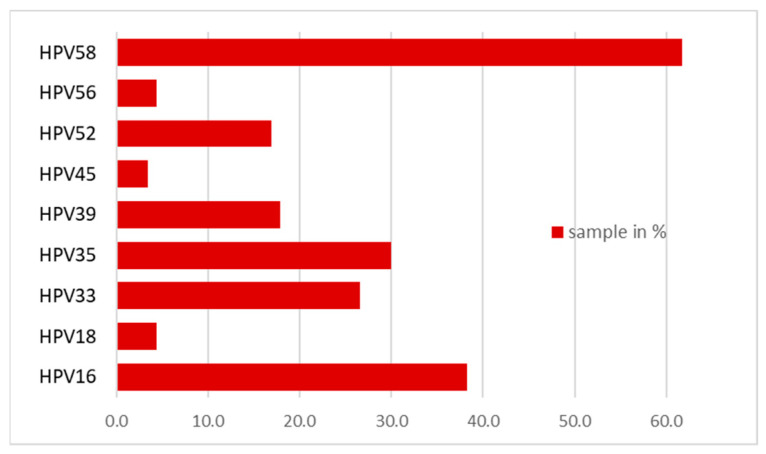
Presence of high-risk HIV genotypes in periodontitis bacteria-positive patients.

**Figure 3 viruses-15-00809-f003:**
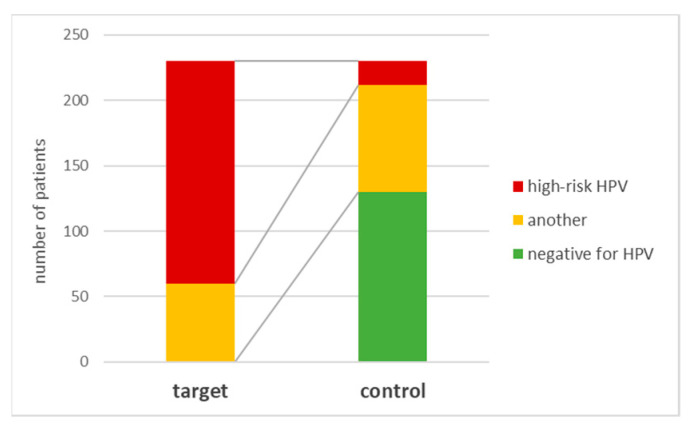
Presence of high-risk HPV genotypes in coronal margin samples of target and control groups.

**Table 1 viruses-15-00809-t001:** Primers used for detection of HPV and the beta globin gene.

Primers	Sequence 5′–3′	Position	Product Size
MY09 *	CGTCCMARRGGAWACTGATC		
		L2 and L1 genes	450 bp
MY 11 *	GCMCAGGGWCATAAYAATGG		
GH20 **	GAAGAGCCAAGGACAGGTAC	Hemoglobin subunit B	268 bp
PC04 **	CAACTTCATCCACGTTCACC		
GP-E6-3F ***	GGGWGKKACTGAAATCGGT		
GP-E7-5B ***	CTGAGCTGTCARNTAATTGCTCA	E6 and E7 genes	650–700 bp
GP-E7-6B ***	TCCTCTGAGTYGYCTAATTGCTC		

* Primers for HPV according to Gravitt et al. [24]; ** primers for beta globin according to Saiki et al. [25] and Ritari et al. [26]; *** primers for HPV according to Sotlar et al. [23].

**Table 2 viruses-15-00809-t002:** Results of HPV screening and genotyping of 230 gingival margin samples positive for periodontitis bacteria.

HPV Genotype	Number	Samples Percentage (%)
6	88	38.3
18	10	4.3
33	61	26.5
35	69	30.0
39	41	17.8
45	8	3.5
52	39	17.0
56	10	4.3
58	142	61.7
* Another	60	26.1
Negative for HPV	0	0

* “Another”: the sample is positive with consensus primers MY09/11 and GPE6/5B/6B but is negative after genotyping by kit for 12 high-risk HPV genotypes 16, 18, 31, 33, 35, 39, 45, 52, 56, 58, 59, and 66. The total of the percentages does not equal 100 because more than 1 HPV genotype was found in 110 samples.

**Table 3 viruses-15-00809-t003:** Results of screening and genotyping for HPV of coronal margin samples of a control group of 230 people.

HPV Genotype	Number	Samples Percentage (%)
18	9	3.9
45	9	3.9
* Another	82	35.7
Negative for HPV	130	56.5

* “Another”: the sample is positive with consensus primers MY09/11 and GPE6/5B/6B but is negative after genotyping by kit for 12 high-risk HPV genotypes 16, 18, 31, 33, 35, 39, 45, 52, 56, 58, 59, and 66.

**Table 4 viruses-15-00809-t004:** Results of screening and genotyping for HPV of 230 buccal mucosa samples of the control group.

HPV Genotype	Number	Samples Percentage (%)
18	16	7.0
* Another	64	27.8
Negative for HPV	150	65.2

* “Another”: the sample is positive with consensus primers MY09/11 and GPE6/5B/6B but is negative after genotyping by kit for 12 high-risk HPV genotypes 16, 18, 31, 33, 35, 39, 45, 52, 56, 58, 59, and 66.

## Data Availability

The data presented in this study can be found in the manuscript and in the Appendix A.

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
