# Peer review of "Molecular Screening for High-Risk Human Papillomaviruses in Patients with Periodontitis"

_viruses, 2023, doi:10.3390/v15030809_

Round 1
Reviewer 1 Report
The work describes an interesting correlation between periodontal disease associated with bacterial infection and the presence of HPY visus in high-risk variants. The sample analyzed is suitable in consideration of the characteristics of a prospective/descriptive mono-institutional study.
The possibility of having rapid probes for PCR typing of the E6 and E7 sequences of the virion allows the identification of high-risk strains of alpha HPV. However, the correlation with a malignant neoplasm at the level of the oral cavity is relatively low, as is the incidence of HPV infection at this level compared to the oropharynx, the authors underline how the anatomical alterations related to periodontal disease can modify the environment where the virus can penetrate and replicate.
The strengths of the work are those of having descriptively investigated a relatively large sample and correlated it with a control group
The critical issues encountered are:
a) the introduction is certainly interesting but contains many known data not related to the scope of the work
b) the inclusion / exclusion criteria of the sample are absent (diabetes, previous dental treatments, age, alcohol abuse, smoking, immunosuppression, etc.)
c) It is not clear what the primary and secondary end points are
d) statistical processing must be completely revised
Consequently, the discussion and the conclusions could support or deny the hypotheses provided by the authors.
The work thus formulated does not meet the standards of the journal and offers a limited contribution to known epidemiological data.
Author Response
We thank the reviewers for their constructive comments, questions and recommendations. We have revised the manuscript according to them.
Find the attached file, please!
Sincerely yours,
The corresponding author
Dr. Ivo Sirakov, PhD

Reviewer 2 Report
The authors have conducted a clinical survey of the prevalence of HPV in patients with periodontal disease in Bulgaria. Its a good sized cohort of 230 patients with double the number of control patients. They report that basically 100% of people with periodontal disease have detectable HPV, with HPV58 being the most frequently detected. The frequency of HPV detection in control patients is statistically lower, at about 35%, and seem to be more related to HPV18 (of the types that can be specifically detected). They seem to conclude that oral HPV infection is likely an opportunistic infection of the inflamed/exposed gum tissues, which seems very likely.
Overall, this is an interesting study. It could be written a little better and is missing some important details. Several other types of analysis may be possible as suggested below.
Major points:
1) The cohort needs to be explicitly described with clinical variables like sex, age, smoking status (if available), HPV vaccine status (if available). This opens the door to additional analyses related to sex and age to see is certain HPV types are more predominant in young vs old or male vs female sex. I would also like to see some evidence that the control cohort has a reasonably similar age and sex distribution to the disease cohort.
2) Its never mentioned which bacteria were detected that are associated with periodontitis. This should be stated. If there are multiple types, are any bacterial species associated specifically with certain HPV types?
3) It is stated that beta-globin DNA was used as a control on line 114. A band of that size is not present on the images shown in figure 1. The authors need to explicitly state that all samples yielded an appropriate control DNA band after PCR, or indicate that if no globin signal was detected, samples were discarded. Personally, it seems slightly unlikely that not a single sample had to be discarded as control DNA was not detected. Usually something goes wrong with the odd sample in a cohort this size.
4) This is not well referenced. There are papers out there describing the relationship between HPV in the oral cavity and cervix in women and men (example PMID: 17021055, PMID: 26503510)
The authors should do a thorough job looking for these types of paper and incorporate them into the intro and discussion. It curious that HPV58 is so common. How prevalent is it in the women of Bulgaria based on previous studies cited on line 54?
There are certainly other papers linking HPV infection to periodontitis (see PMID: 29555599) but they are not cited either. Again, the authors need to dig through pubmed to put their work into perspective with current literature.
Minor Points:
1. Table 1 - not sure what the character that looks like a backwards letter N is between E6 and E7.
2. The first paragraph of the results is just text from the template and should be deleted.
3. there are a few sentences that are missing something. See line 296 for examples, which probably needs something like "the absence of a clinical PATHOLOGY caused by HPV.."
Author Response

(The authors gave the same response as above.)

Round 2
Reviewer 1 Report
The article in the revised format and content corresponds to the needs of a descriptive study on the association of the different forms of periodontal disease and the presence of high-risk HPV. In response to the authors, the need to define the objectives of the study and the stratification of cases based on the most frequent comorbidities arises from the fact that the mechanism of the E6 and E7 sequences of the viral genome can act in symbiosis with environmental factors such as inflammation chronic. However, the work can retain the characteristic defining a demographic area to what extent the presence of periodontal disease can be correlated to HPV infection